# The Development of an In Vitro Horizontal Diffusion Cell to Monitor Nasal Powder Penetration Inline

**DOI:** 10.3390/pharmaceutics13060809

**Published:** 2021-05-28

**Authors:** Péter Gieszinger, Tamás Kiss, Piroska Szabó-Révész, Rita Ambrus

**Affiliations:** Faculty of Pharmacy, Institute of Pharmaceutical Technology and Regulatory Affairs, University of Szeged, H-6720 Szeged, Hungary; gieszinger.peter@pharm.u-szeged.hu (P.G.); tamas.kiss@pharm.u-szeged.hu (T.K.); revesz@pharm.u-szeged.hu (P.S.-R.)

**Keywords:** modified penetration test, nasal powder, levodopa, meloxicam, lamotrigine, diffusion cell, real-time analysis

## Abstract

The development of in vitro investigation models could be important using sensitive and fast methods during formulation. Intranasal applied drugs (meloxicam, lamotrigine, and levodopa) avoid the gastrointestinal tract and can achieve higher bioavailability, therefore a penetration extent is a key property. In this study, the in vitro adaptability of a modified horizontal diffusion cell was tested by using these model active pharmaceutical ingredients (APIs). The special factors consisted of the volume of the chambers, the arrangement of the stirrers, the design of probe input for real-time analysis and decreased membrane area. Membranes were impregnated by isopropyl myristate and by using phosphate buffer to evaluate the effect of API hydrophilicity on the diffusion properties. The lipophilicity of the API was proportional to the penetration extent through isopropyl myristate-impregnated membranes compared with buffer-soaked membranes. After evaluating the arithmetic mean of standard relative deviations and the penetrated extent of APIs at 15 min, Metricel^®^ could be suggested for levodopa and meloxicam, and Whatman™ for lamotrigine. The modified model is suitable for inline, real-time detection, at nasal conditions, using small volumes of phases, impregnated membrane, to monitor the diffusion of the drug and to determine its concentration in the acceptor and donor phases.

## 1. Introduction

Over the last few years, the importance of nasal formulations has significantly improved because intranasal administration can be a non-invasive, alternative choice instead of intravenous—or other intakes—thanks to the quick absorption ability and the rich blood supply [1]. It has a number of advantageous properties, among others quick onset, avoidance of the liver, good compliance, and direct nose-to-brain connection due to the axonal transport [2,3]. However, there are major limiting factors in the application of nasal products, such as the limited residence time on the nasal mucosa caused by mucociliary clearance, the enzymatic barrier, the physical barrier of epithelial cells and the mucous layer [4], therefore, the usual residence time is around 10–15 min, but it can become longer with the use of mucoadhesive materials [5]. The nose comprises two areas remarkable from the absorption aspect. The olfactory region is around 10% of the whole nasal mucosa [6]. A further 90% is made up by the respiratory region, where the API can be absorbed into the blood, after which it can be distributed [7].

APIs can be administered in different dosage forms like solutions, suspensions, emulsions, sprays, nasal drops, nasal powders, nasal gels, or ointments. The optimal nasal formulation has a small volume (25–200 μL), is rapidly absorbable (10–15 min), non-irritating, and simply and accurately usable. Although nasal liquid or gel formulations are present in greater numbers on the market, solid formulations—mainly powders—have been in the focus of scientific attention. Nasal powders are defined as “powders for insufflation into the nasal cavity by means of a suitable device” (Ph. Eur, 9th Ed.). They own several advantageous properties compared to intranasally administered liquid formulations, e.g., prolonged residence time on the nasal mucosa (lower clearance rate and higher mucoadhesivity), higher bioavailability, higher administrable dose, therefore, they can offer a possibility for the non-invasive administration of drugs that have low stability and high dosage, which can lead to higher bioavailability [8], their production can be simpler and reproducible [9,10]. Over recent years, there have been some new nasal powder products on the market [11,12], e.g., BAQSIMI™ nasal powder for the treatment of sever hypoglycemia. In liquid formulations, there are widespread intranasally administered drugs, like vasopressors (xylometazoline and oxymetazoline) or antiallergic agents (mometasone, azelastine, and fluticasone). Additionally, there are special APIs, hormones used by much fewer people (sumatriptan, fentanyl, desmopressin, and calcitonin).

From the formulation aspect, molecular weight, size, solubility, partition coefficient, and pKa value have an effect on the adhesion, liberation and absorption of the nasal formulation. Besides the size of the droplet/particle determines the adhesion properties of the product. Therefore, the proper average particle size can be found in the 5–40 μm range. 

In the case of nasal formulations, the uniformity of mass, the API content, or particle size should be investigated, but the penetration extent is also a key property. Recently, screening methods exist which are able to give information about the adaptability of a nasal pharmaceutical preparation, thereby reducing the risk of human clinical investigations. A general penetration investigation protocol of a pharmaceutical nasal composition is proposed in Figure 1.

The development of in vitro models is exceedingly significant from economic and ecological aspects because if these demonstrate a good correlation with the in vivo studies, pharmaceutical development can be radically accelerated by reducing the time and other sources invested in animal studies [13,14,15,16]. 

In the literature, numerous diffusion apparatuses can be found for penetration studies, the ones with the most widespread use are listed in Table 1.

According to the presented models, the most important parameters are the orientation of the phases, the applied volume, the control of temperature and mixing rate, and the class of process analysis (inline, atline, online, and offline).

The Side-Bi-Side™ (PermeGear Inc., Hellertown, Pennsylvania, United States) can be used—among others—during blood–brain barrier and nasal research, however, the volume of the acceptor and donor compartments are small (3 mL). The two compartments are horizontally located, separated with an artificial membrane (in vitro) or a tissue (ex vivo). It has 3 mL of donor and acceptor phase. Magnetic stirring is possible in both sides, therefore uniform API distribution can be provided. It is proper for the investigation of small amounts of samples. The direction of diffusion can be horizontal and vertical. The sample is applied into the chamber in the case of the horizontal cell and directly onto the membrane surface in case of the vertical cell. It is suitable for the development of suspensions with small volume or diluted macromolecular solutions. However, its modification is necessary and important for the investigation of nasal powders.

Recently, the trends are going in the direction of inline/real-time measurements because they lead to more efficient controllability of the processes. Based on the literature review, an easy-to-use device for measuring the penetration properties of nasal powders is missing from the market, therefore our goal was to make an inline device and test the used parameters. In this work, the adaptability of the modified horizontal cell was investigated as an in vitro method for testing the penetration from nasal powder. The novelty of our modified equipment was given by the horizontal orientation and the novel geometrical developments, the volume of the phases based on the nasal conditions and the inline monitoring of the process, which is useful to test nasal powder products without remarkable extent of aggregation.

## 2. Materials and Methods

### 2.1. Materials

Three types of model APIs were chosen having different lipophilicity. Levodopa (LEV)—logP = −2.39—3,4-dihydroxy-L-phenylalanine was obtained from Hungaropharma Ltd. (BHungary). Meloxicam (MEL)—logP = 3.43—4-hydroxy-2-methyl-N-(5-methyl-2-thiazolyl)- 2H-benzothiazine-3-carboxamide-1,1-dioxide was purchased from EGIS Ltd. (Budapest, Hungary). Lamotrigine (LAM)—logP = 2.57—6-(2,3-Dichlorophenyl)-1,2,4-triazine-3,5-diamine was obtained from Teva Ltd. (Budapest, Hungary).

Three types of membrane potentially suitable for in vitro diffusion investigations were used. Their effect on the penetration of different APIs was evaluated. The chemical structures of the membranes are shown on Figure 2. The Metricel^®^ Membrane Filter (0.45 µm, 25 mm), a mixed cellulose esters membrane (Figure 2A) was purchased from PALL Corporation (Merck, Darmstadt, Germany). Isopore™ Membrane Filter (pore size: 0.40 µm, 25 mm), a polycarbonate membrane (Figure 2B) was obtained from Merck Millipore Ltd. (Cork, Ireland). Whatman™ (0.45 µm, 25 mm), a regenerated cellulose membrane (Figure 2C) was obtained from GE Healthcare Sciences (Chalfont St Giles, UK).

One of the impregnation agents was isopropyl myristate (IPM) obtained from Sigma Aldrich (Budapest, Hungary), the other one was pH = 7.4 phosphate buffer as a reference. The effect of IPM on the penetration properties was evaluated.

### 2.2. Particle Size Determination of the APIs

The particle size distribution of the powders was estimated by laser diffraction (Malvern Mastersizer Scirocco 2000, Malvern Instruments Ltd., Worcestershire, UK). The average particle size was characterized by the d(0.5) value. In the dry analysis method, approximately 1.0 g of product was loaded into a feeding tray. Three parallel measurements were carried out for every API. The dispersion air pressure was adjusted to 2.0 bar in order to determine whether particle attrition had occurred. Obscuration between 10.0% and 13.0% was achieved throughout the entire duration of the measurement. Three repeat measurements were performed on each sample and the mean value was calculated. The residual value was always <1.0%. The products in the vacuum cleaner were collected for further studies when the dry analysis was completed. Air was used as the dispersion medium for the ground products from the entrance to the sample cell. The d(0.5) value of particles in the nasal powders is preferred to be in the 5–40 μm diameter range as these particles can adhere in the upper airways with the best efficiency [25].

### 2.3. Hydrophilicity Determination of the Membranes with Contact Angle Measurements

The contact angle (*θ*) was measured, using an OCA 20 Optical Contact Angle Measuring System (Dataphysics, Filderstadt, Germany) and Wu’s method for determination of surface energy and polarity of membranes. Then, 3 cm^2^ of the membrane was placed onto a glass plate. The contact angle of water and diiodomethane was measured. Using the following Wu’s equation, the surface energy of the membranes (*γ^*^*) could be determined:$$\left(1-\cos\theta \right)=\frac{4\gamma_{s}^{d}\gamma_{l}^{d}}{\gamma_{s}^{d}+\gamma_{l}^{d}}+\frac{4\gamma_{s}^{p}\gamma_{l}^{p}}{\gamma_{s}^{p}+\gamma_{l}^{p}}$$

*θ* = contact angle, *γ_s_^d^* = dispersive component of solid material surface energy, *γ_s_^p^* = polar component of solid material surface energy, *γ_l_^d^* = dispersive component of liquid surface tension, and *γ_l_^p^* = polar component of liquid surface tension.

When dropping, in 1–25 s time interval, the contact angle change was measured (1 Hz sampling frequency), therefore the change of contact angle was followed.

At the same time as the dropping, we made a recording by setting the apparatus to 1–25 s time interval; thereby, the detection and determination of the change of the contact angle were possible. Thus, we obtained the contact angle—always in the same second—of the two different applied fluids.

It is also worth mentioning that the surface energy of the solid material is the sum of polar and dispersive surface energy: *γ^*^* = *γ_l_^d^* + *γ_l_^p^*.

The surface tension of polar component was *γ^p^* = 50.2 mN/m, interfacial tension of dispersive component was *γ^d^* = 22.6 mN/m and diiodomethane (*γ^p^* = 1.8 mN/m, *γ^d^* = 49 mN/m). 

This equation can be written with both liquids and the 2 unknown members (*γ_l_^d^*, *γ_l_^p^*) can be defined solving the system of equations.

The polarity of the solid material can be expressed by dividing the polar component of free energy by free energy:$$Polarity\;\left(\%\right)=\frac{\gamma_{s}^{p}}{\gamma^{*}}*100$$

### 2.4. Factors Influencing the Results of the Diffusion Measurements

For this identification, a knowledge space development was executed as part of the Quality by Design methodology and an Ishikawa diagram was therefore set up. With the Ishikawa diagram, the identification of influencing factors was carried out, then the most significant ones were chosen and varied. After the most significant factors were identified, an experimental design was set up and a horizontal cell method was modified.

### 2.5. In Vitro Diffusion Measurement by the Horizontal Cells

At the beginning of the measurements, 4.5 mL of so-called simulated nasal electrolyte solution (SNES) [26,27] was added to the donor chamber, then the complete amount of API powder was washed with another 4.5-mL portion. It was the zero moment of the penetration measurements. The acceptor phase was adjusted with pH = 7.4 ± 0.1 phosphate buffer solution (PBS) imitating the conditions of the blood. The composition of PBS was: 10.0 g/L of NaCl, 0.20 g/L of KCl, 1.15 g/L of Na_2_HPO_4_ and 0.12 g/L of KH_2_PO_4_. The SNES consisted of 8.77 g of NaCl, 2.98 g of KCl, and 0.59 g of anhydrous CaCl_2_ in 1000 mL of deionized water, the pH was set at 6.0 ± 0.1 with HCl. The system was thermostated with the circulation of deionized water at 32 °C. NaCl, KCl, CaCl_2_, Na_2_HPO_4_, and KH_2_PO_4_ were purchased from Sigma-Aldrich (Budapest, Hungary). The rotational speed of the magnetic stirrers was around 100 rpm. The membranes were soaked into the impregnation agent for 30 min before every investigation.

A mass of 10 ± 0.50 mg of powder of the APIs was weighed and the results were corrected depending on the actual weight because the human nose can accommodate around 10–25 mg of powder per nostril per shot [28]. The penetration extent (P_API_, μg/cm^2^) can be used in the case of nasal formulations to describe the extent of diffusion. In spite of the fact that, thanks to mucociliary clearance, the residence time of nasal formulations is around 15 min on the nasal mucosa, the measurements lasted for 60 min so that the kinetics could be better characterized. The penetration extent at 15 min was used to define the sequence among the APIs diffused.

The diffusion surface was 0.785 cm^2^. The area of the nasal mucosa in the human body is around 200-times larger: 160 cm^2^ [29], therefore presumably multiple times more API is supposed to be penetrated in vivo than in the horizontal device.
$$P_{\mathrm{API}}=\frac{\mathrm{Mass}\;\mathrm{of}\;\mathrm{penetrated}\;\mathrm{API}\;\left(µg\right)}{\mathrm{Diffusion}\;\mathrm{surface}\;\left({\mathrm{cm}}^{2}\right)}$$

Three parallel measurements were carried out with every composition of the system to get information about the precision. The device was connected to a probe suitable for the real-time detection of the penetrated API content. 

### 2.6. Offline Spectrophotometric Measurements

At determined times (5, 10, 15, 30, 45, and 60 min), 2.0 mL samples were taken from the acceptor phase and the volume was completed to 9.0 mL, so the acceptor phase was diluted after every occasion. The dissolved drug amount was determined spectrophotometrically (PerkinElmer, Lambda 20 spectrophotometer, Dreieich, Germany). The optical path length was 1 cm.

MEL, LAM, and LEV were quantified at wavelengths of 366, 307, and 281 nm, respectively.

### 2.7. Inline Measurements with a Probe Connected to a Spectrophotometer with Optical Fiber

An AvaLight DH-S-BAL spectrophotometer (Avantes, Apeldoorn, The Netherland) was connected to an AvaSpec-2048L transmission immersion probe (Avantes, Apeldoorn, The Netherland) with optical fiber to quantify the amount of API. The optical path length was 1 cm. The limiting factor of inline measurements can be that no dilution occurs in the acceptor phase during the measurements which could model the dilution happening in the blood by transferring the API from the direct environment of the nose [30]. The standard deviation of the layouts was calculated based on the 3 parallel measurements made at the same times (5, 10, 15, 30, 45, and 60 min) as during the offline investigations.

MEL, LAM, and LEV were quantified at wavelengths of 366, 307, and 281 nm, respectively.

## 3. Results and Discussion

### 3.1. Evaluation of the Factors Affecting the Adaptability of the Horizontal Diffusion Cells for the Penetration Measurements of Nasal Powders

Figure 3 shows the Ishikawa diagram of the robustness of this kind of quantification method. It can be seen that there are five groups of influencing factors: device, liquid medium, API quantification, implementation, and the properties of the API. 

The deep review of the literature revealed that the types of membrane, the impregnation agent, the quantification method and the logP value of the API seemed to be the most influencing factors during the diffusion measurements, by varying these values, effect was evaluated on the results. As the most significant factors were identified, an experimental design was constructed.

### 3.2. Measurement of the Size Distribution of Applied Model APIs

The d(0.5) value of the APIs is in the preferred 5–40 μm range (Table 2), therefore they are proper for modeling a nasal powder. The d(0.5) value of all powders satisfied the particle size distribution requirements of nasal powders.

### 3.3. Determination of Surface Free Energy and Polarity of Membranes Using Wu’s Method

Based on the results of the contact angle measurements with water and diiodomethane, a hydrophobicity sequence was set up (Table 3).

The hydrophobicity of the membranes was proportional to the hydrophobicity of the APIs, mainly in the case of the reference pH = 7.4 phosphate buffer impregnation because the membrane influences the results, while in the case of IPM impregnation, the impregnation layer thickness also depends on the hydrophobicity of the membrane, thereby affecting the results. There are two similar membranes (Metricel^®^ and Isopore™) and a more polar membrane (Whatman™).

### 3.4. The Design of the Horizontal Diffusion Cell Appropriate for the Investigation of Nasal Powders

The geometry of Side-Bi-Side™ apparatus was reconsidered and a novel implementation was optimized for nasal powder penetration investigations (Table 4). 

A horizontal diffusion device (Figure 4) was tested and used to optimize a method proper for API investigations comprising different logP values. The factors listed on the Ishikawa diagram influence the efficiency of in vitro membrane diffusion to a different extent. 

This device consists of two chambers: a donor and an acceptor phase with a horizontal orientation. The volume of the donor and acceptor phases was 9 mL, and a membrane surface with an area of 0.785 cm^2^ was used. The volume of the nasal cavity is 15–20 mL, which is divided into fossae by the nasal septum, therefore 9 mL was the ideal choice to model the absorption in the nasal fossa. Besides the design of the chambers was constructed to make the space suitable for real-time analysis with an immersion probe input. The equipment was thermostated by a water jacket with the help of a circulator. The temperature was set at 32 °C, which is the usual temperature inside the human nose. These key conditions were necessary to be controlled so that the results could be reproducible. Magnetic stirrers were located under the liquid in a specially formed hollow and were used to achieve the homogeneity in the solutions and to imitate the intensive air circulation of the nose. The magnetic stirring bars were moved by CS-DSD1 Digital Magnetic Stirrer with 2 × 6 position (CS-Smartlab Devices Ltd., Kozármisleny, Hungary). The purpose of these changes was to model the nasal environment with better efficiency and to achieve more precise monitoring of the process. The powders could be homogeneously suspended in the donor phase which is a condition of the reproducible measurements.

The in vitro–in vivo correlation (IVIVC) was investigated, the data of the measurements detailed presented in this article and Side-Bi-Side™ data were compared to in vivo results which proved that the offline modified cell was more closely-correlated (R = 0.9580) with the in vivo data than the Side-Bi-Side™ apparatus (0.9532). In addition, the inline monitored, modified cell performance (R = 0.9727) correlated the most closely with the in vivo results. The data of Side-Bi-Side™ and the in vivo results were collected from previous articles of our research groups [31,32]. This statistical comparison was based on the AUC values between the determined sampling points from the apparatus correlated with in vivo AUC values between the same sampling points.

In the in vivo studies, suspended MEL was intranasally administered for rats. A 1 mg/mL MEL containing suspension was prepared comprising 5 mg/mL sodium-hyaluronate excipient which was definitely necessary to provide viscosity for the formulation in the nose. In the compared measurements, Metricel^®^ membrane was used. Due to the close correlation between the in vivo and the in vitro data, it was concluded that the modified inline apparatus was adaptable for nasal powder penetration studies. 

### 3.5. Determination of Calibration Curves by Offline and Inline Monitoring

The calibration equations were calculated in PBS as the acceptor phase. The linearity range of the calibration curve: the limit of detection (LOD), the lower limit of quantification (LLOQ), and the upper limit of quantification (ULOQ) were defined with the inline and offline setup as well. The x values are in μg/mL unit, the y values are in A.U. The analytical parameters of the measurements are listed in Table 5.

### 3.6. In Vitro Diffusion Studies

Two kinds of data were used to determine the in vitro modeling efficiency: the penetration extent at 15 min and the arithmetic mean of the standard relative deviations.

MEL penetration through the membrane occurred based on the high lipophilicity of MEL (Figure 5A,B).

As it is poorly water-soluble (with the highest logP value among the model APIs), IPM impregnation affected its ability to diffuse to the acceptor phase with the highest efficiency compared to soaking with PBS. It tended to cross the IPM-impregnated membrane, thus causing a higher diffusion rate and API concentration in the acceptor phase (Table 6).

The inline measurements were more precise than the offline ones (lower relative standard deviation values), therefore the inline ones should be preferred. IPM-impregnated Metricel^®^ was the most effective membrane in inline measurements based on the high extent of penetration (Figure 5 and Table 6) and the low relative standard deviation (Table 7) among all setups. This membrane might function properly because of its apolar property.

LAM has moderate hydrophobicity; accordingly, its penetration properties are between those of LEV and MEL. There was a smaller difference between the penetration extent of impregnation with the hydrophilic PBS and the hydrophobic IPM than in the case of MEL. The highest rate of penetration among them belonged to the Whatman™ membrane (Figure 5C,D).

The phenomenon of high penetration rate through the IPM-impregnated Whatman™ membrane could occur due to the low hydrophobicity of Whatman**™** and the high hydrophobicity of IPM, which could compensate the effect of each other to achieve moderate impregnated membrane hydrophobicity, which was advantageous for the moderately hydrophobic LAM.

In most setups, the lowest relative standard deviation belonged to the inline measurements with IPM impregnation. When soaking in PBS, offline measurements showed a lower relative standard deviation but when using the IPM, the tendency is not so obvious. The arithmetic mean of the standard deviations of Metricel^®^ membrane was a bit lower compared to Whatman™, the higher extent of penetration was in favor of the use of Whatman™ in inline measurements when the impregnation agent was IPM. In the case of IPM impregnation, the inline measurements were more precise than the offline ones (Table 7), therefore they are suggested. 

LEV could diffuse across the membrane with the highest rate among the investigated APIs when penetration was carried out in PBS versus IPM soaking (Figure 5E,F). It was because of the high hydrophilicity of the API. 

The penetration extent was similar in inline measurements when the membranes were impregnated with IPM. Isopore™ membrane impregnated with IPM could not be used due to the low extent of penetrated LEV. In the offline method, Metricel^®^ was considered to be the most precise method in inline and offline cases if soaked in IPM (Table 7).

The IPM-impregnated Whatman™ behaved comparably to Metricel^®^, but the arithmetic mean of its relative standard deviation was higher (Table 7), therefore Metricel^®^ was suggested with IPM impregnation for the in vitro investigation of LEV containing nasal powders with inline detection.

## 4. Conclusions

It can be stated that the inline results show similarity to the offline cases. Hydrophobic MEL penetrated with the highest efficiency through the membrane, the penetration was poorer in the case of LAM, and even poorer for LEV when the impregnation agent was hydrophobic IPM. When soaking in PBS, the tendency was reversed.

In this study, we aimed to develop and validate an inline horizontal penetration cell for three model APIs with different lipophilicity. This equipment was constructed by the modification of a Side-Bi-Side™ cell, the results more closely correlated with the offline and the inline results of the modified cell than the original apparatus. Besides it was also an important aspect that it was also proper for inline measurements in contrast of the Side-Bi-Side™ because the immersion probe had enough space in the acceptor phase with increased volume. To optimize the setup further and to analyze and interpret the results, three types of API, membrane, two impregnation methods, and two analytical instruments were used, based on a prior knowledge space development with the help of an Ishikawa diagram. In terms of the analytical methods, the results were similar in both cases, but the investigations with the UV-Vis probe method were easier to carry out because the possibility of human mistakes is lower, the error of dilutions and samplings does not exist. In most cases, this led to lower standard deviations. It is also advantageous that the result can be detected real time, so we can get information about the kinetics of the experiment immediately, and if any problems are detected, the measurements can be interrupted or aborted. In the inline in vitro nasal diffusion experiments, Metricel^®^ is suggested for the investigation of MEL and LEV, and Whatman™ for the investigation of LAM with 0.5 h of IPM impregnation of the membrane. This conclusion was drawn based on that the API permeation through the ideal membrane is high and the arithmetic mean of relative standard deviation is low. Based on this aspects, the Metricel^®^ membrane seemed to be the ideal choice for MEL and LEV, Whatman™ was optimal in the case of LAM, when the results were monitored with an immersion probe. The IPM impregnation was indispensable because it provides the lipophilicity of membrane modelling the nasal mucosa lipophilicity, the pH = 7.4 impregnation was used as a reference to evaluate the effect of IPM impregnation. The IPM impregnation had a diffusion-promoting effect when the API logP was high. However, the in case of APIs with low logP, it did not help the diffusion because an API with low lipophilicity could not penetrate through a lipophilic-impregnated membrane. This phenomenon definitely occurs because the lipophilic API is able to cross the hydrophobic IPM, causing higher concentration in the acceptor phase. Although the penetrated concentrations of APIs seem to be low, taking into consideration the fact that the surface of the nasal mucosa is around 200-times larger than the in vitro surface in the setups, the penetration is supposed to be significantly higher in the human nose. During our work, a horizontal in vitro inline setup was successfully developed to measure the penetration of an increasingly popular formulation, the nasal powder, in the case of model APIs, with different hydrophobicity.

## Figures and Tables

**Figure 1 pharmaceutics-13-00809-f001:**
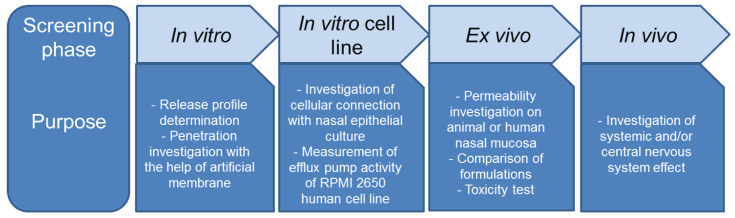
Suggested investigation protocol during formulation development of nasal formulations.

**Figure 2 pharmaceutics-13-00809-f002:**
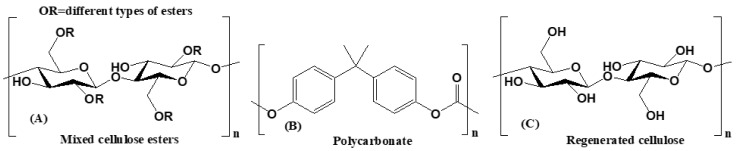
The chemical structure of the membranes: (**A**) Metricel^®^, (**B**) Isopore™, and (**C**) Whatman™.

**Figure 3 pharmaceutics-13-00809-f003:**
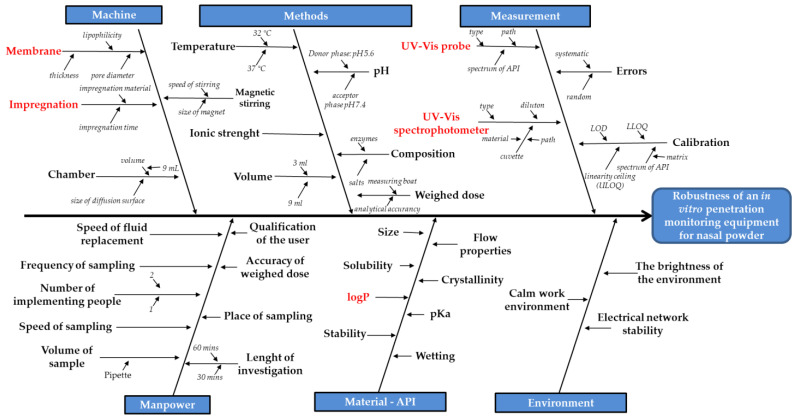
Ishikawa diagram of developing a system for in vitro penetration measurements of nasal powders, the most influencing factors are highlighted in red.

**Figure 4 pharmaceutics-13-00809-f004:**
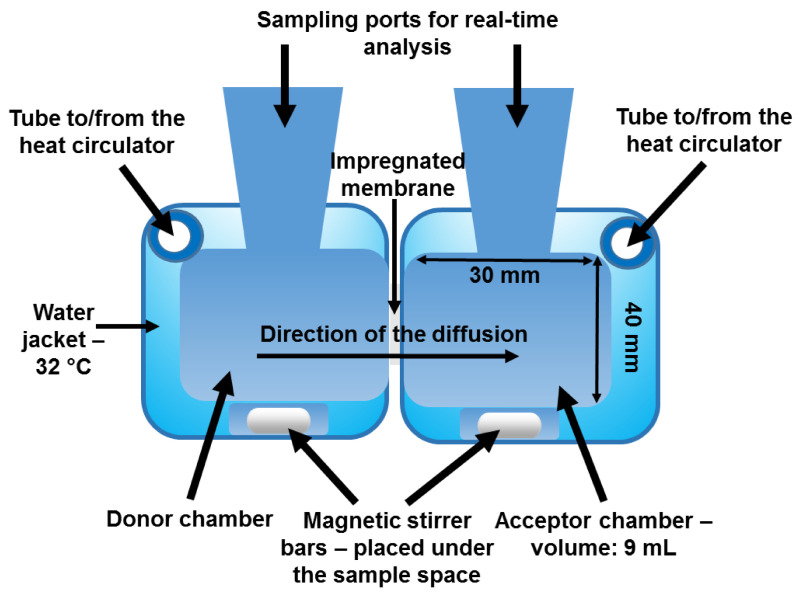
The setup of the modified horizontal device used for in vitro modeling of the penetration of nasal powders—the specified features of the different parts are described.

**Figure 5 pharmaceutics-13-00809-f005:**
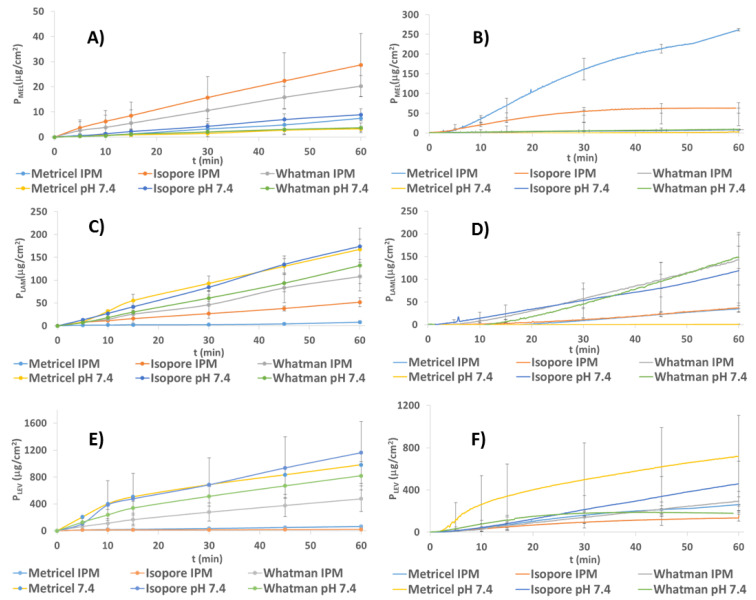
Penetration of MEL through different membranes during offline (**A**) and inline measurements (**B**), LAM during offline (**C**) and inline measurements (**D**), and LEV during offline (**E**) and inline measurements (**F**) via soaking in PBS or in IPM.

**Table 1 pharmaceutics-13-00809-t001:** The most commonly used in vitro diffusion investigation devices.

Name	Application	Direction of Diffusion	Volume and Measurement Conditions
μFLUX™ diffusion cell—in situ UV monitoring optical system (Pion Inc., Billerica, Massachusetts, United States) [17]	Diffusion measurement with artificial membrane	Horizontal	Both phases: 10–13 mL Real-time analysis is possible
Navicyte Vertical Diffusion Chamber System (Harvard Apparatus, Holliston, Massachusetts, United States)	Cells, ex vivo tissue investigation Intestinal and nasal permeability	Vertical	Both phases: variable Real-time analysis is possible
Navicyte Horizontal Diffusion Chamber System (Harvard Apparatus, Holliston, Massachusetts, United States) [18,19]	Tissues contacting with air (nasal, pulmonary, dermal) permeability	Horizontal	Both phases: variable Real-time measurement is possible
In-Line Cell (PermeGear Inc., Hellertown, Pennsylvania, United States) [20,21]	Permeability of transdermal and buccal formulations	Vertical	Acceptor phase: variable Donor phase: 100 mL Real-time analysis is possible
Franz vertical diffusion cell (Hanson Research, Chatsworth, California, USA) [16,22]	Permeability of transdermal formulations	Vertical	Donor phase: 300 µL Acceptor phase: 7 mL No real-time analysis
Side-Bi-Side™ horizontal diffusion cell (PermeGear Inc., Hellertown, Pennsylvania, United States) [23,24]	Blood–brain barrier investigations Permeability of nasal formulations	Horizontal	Both phases: 3 mL Real-time analysis is possible

**Table 2 pharmaceutics-13-00809-t002:** Size distribution of the APIs (μm).

API	d (0.1)	d(0.5)	d(0.9)
MEL	2.99 ± 0.17	10.76 ± 0.66	31.72 ± 2.93
LAM	2.82 ± 0.05	11.71 ± 0.41	63.37 ± 7.69
LEV	4.84 ± 0.02	22.25 ± 0.24	81.53 ± 3.28

**Table 3 pharmaceutics-13-00809-t003:** Contact angles, surface free energy, and polarity of membranes.

Membrane	θ _water_ (°)	θ _diiodomethane_ (°)	*γ* (mN/m)	*γ^p^* (mN/m)	*γ^d^* (mN/m)	Polarity (%)
Metricel^®^	29.3	3.4	76.23	30.41	45.82	39.89
Isopore™	40	8.1	71.3	25.8	45.49	36.19
Whatman™	16.2	14.5	79.6	35.24	44.36	44.27

**Table 4 pharmaceutics-13-00809-t004:** The design of the horizontal diffusion cell to optimize it for inline nasal powder penetration measurements.

Change in Design	The Design of Side-Bi-Side™ Horizontal Diffusion Cell	The Design of New Horizontal Cell
Cell volume	3 mL	9 mL
Magnetic stirring bars	In the sample space	In the hollow under the sample space
Sampling ports	Too narrow sampling port for probe input	The design of sampling port for probe input

**Table 5 pharmaceutics-13-00809-t005:** Analytical parameters of the measurements.

API	Offline Measurements			Inline Measurements		
	Calibration Equation	LOD/LOQ (μg/mL)	ULOQ (μg/mL)	Calibration Equation	LOD/ LOQ (μg/mL)	ULOQ (μg/mL)
MEL	y = 0.04849x	0.05296/0.1605	39.09	y = 0.04658x	0.08726/0.2644	40.00
LAM	y = 0.02745x	0.09062/0.2746	20.03	y = 0.02708x	0.05602/0.1698	20.62
LEV	y = 0.01350x	1.736/5.262	90.41	y = 0.01349x	0.2265/0.6863	100.04

**Table 6 pharmaceutics-13-00809-t006:** Penetration extent (μg/cm^2^) of MEL, LAM, and LEV at 15 min of measurements.

API	Connectivity	Offline			Inline		
	Impregnation Agent/Membrane	Metricel^®^	Isopore™	Whatman™	Metricel^®^	Isopore™	Whatman™
MEL	pH = 7.4	0.91	2.20	1.18	−0.37	2.44	3.12
	IPM	1.25	8.54	5.51	69.41	32.00	39.62
LAM	pH = 7.4	55.21	41.73	30.17	0.18	53.48	45.20
	IPM	2.38	15.84	25.56	8.98	11.12	57.04
LEV	pH = 7.4	506.16	476.67	336.98	341.86	81.81	115.60
	IPM	20.70	13.44	165.59	66.20	50.21	63.19

**Table 7 pharmaceutics-13-00809-t007:** Arithmetic mean (%) of the relative standard deviations of MEL, LAM, and LEV during the measurements.

API	Connectivity	Offline			Inline		
	Impregnation Agent/Membrane	Metricel^®^	Isopore™	Whatman™	Metricel^®^	Isopore™	Whatman™
MEL	pH = 7.4	81.44	40.53	93.00	84.53	21.27	62.97
	IPM	66.33	57.79	78.98	14.72	22.86	38.77
LAM	pH = 7.4	17.00	63.02	22.09	89.23	76.39	35.92
	IPM	58.66	14.15	54.90	16.67	27.67	19.54
LEV	pH = 7.4	24.38	12.11	53.73	83.14	66.42	26.84
	IPM	8.57	66.41	31.2	12.36	65.54	51.21

## Data Availability

The data presented in this study are available on request from the corresponding author.

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
