# Peer review of "The Development of an In Vitro Horizontal Diffusion Cell to Monitor Nasal Powder Penetration Inline"

_pharmaceutics, 2021, doi:10.3390/pharmaceutics13060809_

Round 1

Reviewer 1 Report

The author has revised the manuscript according to the reviewer's comments. The current form can be accepted for publication.

Reviewer 2 Report

Dear authors,

Thank you for revising your manuscript and addressing our comments and suggestions.

Now, I think your manuscript has improved remarkably and I have no doubt that it will attract a great interest from the scientific community.

With best wishes  

This manuscript is a resubmission of an earlier submission. The following is a list of the peer review reports and author responses from that submission.

Round 1

Reviewer 1 Report

Dear authors,

Thank you for submitting your manuscript for publication in the Pharmaceutics Journal.

The work is interesting and worth publication. However, extensive editing is required beforehand.

The introduction is too long and very confusing and it is very difficult for the reader to follow and clearly figure the research problem until the last paragraph. Therefore, this should be edited and focus on the current in vitro testing methods for testing the intranasal drug powder delivery and their limitations, which you are trying to address in your research.

The same would be applied to the results and discussion sections.

Good luck.

Reviewer 2 Report

The purpose of the study is to establish an in vitro investigation model for screening of intranasal formulations. The setup was modified from a Side-Bi-Side device. The special factors consisted of the volume of the chambers, the arrangement of the stirrers, the design of probe input for real-time analysis and decreased membrane area. The membrane penetration of meloxicam, lamotrigine and levodopa was measured by using the modified horizontal diffusion cell. I agree with the authors that sensitive and fast in vitro models are important for formulation screening. However, the setups of the new device were not validated.

  1. The horizontal diffusion cell was developed from the commercial Side-Bi-Side cell. The modification consisted of the change in the volume of the chambers (from 3 mL to 9 mL), placing the magnetic stirring bars under the sample space, the design of probe input for real-time analysis, membrane surface with a small area (0.785 cm2). However, all of these changes were not validated with experiments. The authors should provide data to support the optimization of these parameters.

  1. The rationality of the horizontal diffusion cell is of concern. In this setup, the tested drug powders are suspended in the donor chamber, in which only a part of the powders may contact the membrane. The situation is obviously different to the actual situation of the formulation in the nasal cavity, because the nasal powder may be embedded in the mucus and have a contact with the epithelia. It is critical to compare the effects of the direction of diffusion on the adaptability of the model.

  1. The rationality of the membrane should be validated. Since the measurements only lasted for 60 min, it is preferable to adopt nasal mucosa instead of a polymer membrane.

  1. As shown in Figure 5, the results obtained from offline are quite different to that from inline measurements. Reasons for this difference should be discussed. The data from both measurements should be verified.

  1. Metricel® is suggested for the investigation of MEL and LEV, and Whatman™ for the investigation of LAM with 0.5 h of IPM impregnation of the membrane. What are the reasons for this conclusion?

  1. Besides the powders, the adaptability of the device should be tested with other dosage forms such as solutions, suspensions, emulsions, sprays, nasal drops, nasal gels, and ointments.

  1. The “hydrophilicity determination of the membranes” was not clearly described. In addition, what is γp, γd, and γ*?